# Mind-the-Glitch:
# Visual Correspondence for Detecting Inconsistencies in Subject-Driven Generation

**Abdelrahman Eldesokey**   **Aleksandar Cvejic**   **Bernard Ghanem**   **Peter Wonka**
KAUST, Saudi Arabia
`first.last@kaust.edu.sa`

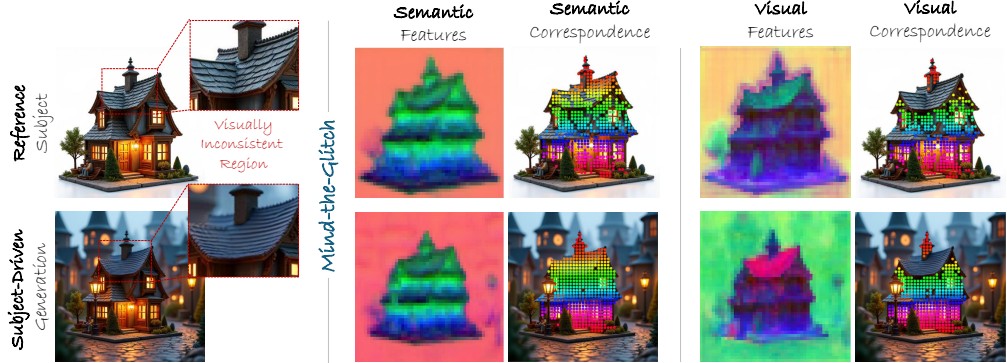

Figure 1: **Mind-the-Glitch** is the first pipeline that enables computing *visual correspondences* based on the backbone features of pre-trained diffusion models. The pipeline separates backbone features into semantic and visual components, allowing for *visually matching* keypoints across images, analogous to the well-established *semantic correspondence* task. This provides the first empirical framework for evaluating and localizing visual inconsistencies in subject-driven image generation.

## Abstract

We propose a novel approach for disentangling visual and semantic features from the backbones of pre-trained diffusion models, enabling *visual correspondence* in a manner analogous to the well-established *semantic correspondence*. While diffusion model backbones are known to encode semantically rich features, they must also contain visual features to support their image synthesis capabilities. However, isolating these visual features is challenging due to the absence of annotated datasets. To address this, we introduce an automated pipeline that constructs image pairs with annotated semantic and visual correspondences based on existing subject-driven image generation datasets, and design a contrastive architecture to separate the two feature types. Leveraging the disentangled representations, we propose a new metric, *Visual Semantic Matching (VSM)*, that quantifies visual inconsistencies in subject-driven image generation. Empirical results show that our approach outperforms global feature-based metrics such as CLIP, DINO, and vision–language models in quantifying visual inconsistencies while also enabling spatial localization of inconsistent regions. To our knowledge, this is the first method that supports both quantification and localization of inconsistencies in subject-driven generation, offering a valuable tool for advancing this task. Project Page: https://abdo-eldesokey.github.io/mind-the-glitch/

39th Conference on Neural Information Processing Systems (NeurIPS 2025).

# 1 Introduction

Recent advances in diffusion models have transformed the landscape of image generation, delivering exceptional quality, fidelity, and diversity [8, 5, 32, 36, 27, 30]. These capabilities have revolutionized several domains, including artistic creation, advertising, content production for video games, and storyboards for movies. In many of these applications, maintaining visual consistency of a subject, whether a character or an object, across different generations is crucial. For example, in cinematic and advertising workflows, preserving the identity and appearance of a subject across multiple scenes is essential for narrative coherence and branding. However, most diffusion models operate in a latent space where semantic and visual concepts are entangled, making it challenging to control or preserve specific content using text prompts alone.

To address these limitations, a growing research direction has focused on *subject-driven* generation, which aims to guide diffusion models to produce consistent images of a given subject in different scenes while preserving fine-grained visual details [13, 33, 28, 3, 49]. Nonetheless, a major bottleneck in this line of research has been the lack of reliable evaluation metrics for subject consistency. Since subjects may appear in varying poses and spatial configurations, traditional image similarity metrics such as LPIPS [53], or structural similarity (SSIM) are not well-suited for this task. As a result, many works have relied on feature-based similarities using models like CLIP [29] or DINO [4] as approximations of overall appearance. However, these metrics capture global semantics and tend to overlook subtle visual inconsistencies in object details [26].

Recent works [41, 26] have explored the use of Vision-Language Models (VLMs), such as ChatGPT, to assess consistency between a reference and a generated image. While these approaches are promising, they remain limited to global assessments, and it is often unclear which visual cues or criteria the models rely on to judge consistency. Furthermore, VLM-based metrics lack the ability to localize inconsistent regions within the subject. To enable robust evaluation of subject-driven image generation, it is essential to develop methods that can reliably match the visual appearance of a subject across images, irrespective of variations in pose, scale, or context, and accurately localize inconsistencies for potential correction through post-processing.

This challenge bears similarities to the task of *semantic correspondence*, where corresponding points are matched across image pairs of objects with varying poses, scales, and contexts. Diffusion models have been shown to encode semantically-rich features that have demonstrated great success in computing semantic correspondences across images [39, 42, 52, 22, 51]. However, as illustrated in Figure 1, these semantic features are typically insensitive to appearance-level variations, as they focus primarily on structural or categorical semantics rather than fine-grained visual details. Given that diffusion models are trained for image synthesis, it is reasonable to assume that their internal representations should encode both semantic and visual information, yet only the semantic component has been extensively leveraged so far.

Building on this hypothesis, we propose a novel framework for disentangling semantic and visual features from the backbone of pre-trained diffusion models. Due to the absence of datasets with annotated visually similar or dissimilar regions of a given subject, we introduce an automated dataset generation pipeline that constructs image pairs with annotated semantic and visual correspondences, derived from existing subject-driven generation datasets. Using this dataset, we propose an architecture that disentangles semantic and visual features in a contrastive manner. We then leverage the disentangled representations to derive a metric, the *Visual Semantic Matching (VSM)* metric, that quantifies the degree of visual inconsistency in subject-driven generation. Empirical results show that our approach outperforms existing feature-based metrics such as CLIP and DINO, as well as the Vision-Language Model (VLM) metric, in quantifying visual inconsistencies, while allowing for localizing the inconsistent regions as well. We believe our framework offers a valuable step forward in the evaluation and development of subject-driven image generation.

Our contributions can be summarized as follows:

- We propose a novel framework comprising an automated dataset generation pipeline and an architecture for disentangling semantic and visual features in diffusion models.
- Based on the disentangled features, we present a new metric for evaluating subject-driven generation methods that both quantifies and localizes visual inconsistencies.
- We empirically demonstrate that the proposed metric outperforms existing feature-based metrics commonly used in subject-driven generation.

# 2 Related Work

In this section, we first provide a brief overview of the *semantic correspondence* task, which serves as a key inspiration for our work. Then we review existing subject-driven image generation approaches, and how consistency is currently evaluated in these methods.

## 2.1 Semantic Correspondence using Diffusion Models

Diffusion models have been shown to encode semantically rich features within their backbones, enabling computing semantic correspondence across instances of the same object class and even among semantically similar categories. Some approaches leveraged intermediate features from the decoder of diffusion models [42], and others combine them with features from DINO [4] to enhance semantic representation [52]. Several works [22, 51] proposed learnable aggregation networks that automatically select and fuse features from different layers, trained in a supervised manner using semantic correspondence datasets such as SPair-71k [24]. CleanDIFT [39] further introduces a distillation framework to compress diffusion features, allowing for efficient inference. These semantically rich features have been successfully applied to a range of downstream tasks, including segmentation [45, 23], controllable editing [25, 2, 7], and object manipulation [20]. Inspired by the task of semantic correspondence, we aim to learn feature representations that capture visual appearance rather than semantics. These visual features enable matching regions based on appearance and are well-suited for detecting visually inconsistent regions in subject-driven image generation.

## 2.2 Subject-Driven Image Generation

Early personalized generation techniques using UNet-based diffusion models [32, 27] primarily focused on encoding subject identity through full-model fine-tuning [34, 17] or low-rank adaptation methods [12, 10, 37, 19], and even training-free approaches [43] Other approaches [35, 9, 15, 1] learned subject-specific embeddings, without modifying model weights, by associating subjects with new tokens in an image or text embedding space. To improve the trade-off between efficiency and fidelity, adapter-based methods [48, 44, 13, 28, 50] introduced zero-shot personalization by conditioning on reference image features through lightweight network modules. These methods showed that injecting image-driven signals or selectively updating parameters could effectively preserve subject identity and visual details, even with limited data.

The emergence of diffusion Transformers (DiTs) [8, 5] introduced a new class of architectures that replace UNets with Vision Transformers as the denoising backbone, offering greater scalability and enhanced contextual understanding. These models support more flexible conditioning mechanisms and exhibit strong in-context learning capabilities, enabling them to generate diverse images of a subject without explicit fine-tuning [38, 18]. To further improve fidelity and identity preservation, several works have applied LoRA-based fine-tuning to DiTs architectures [41, 54, 14, 3], achieving more precise control over subject appearance and consistency. More recently, visual autoregressive models (VARs) have been adopted for subject-driven generation due to their inherently strong conditional modeling capabilities [40, 47, 6]. For an exhaustive overview of subject-driven image generation approaches, we refer the readers to [46].

## 2.3 Evaluating Subject-Driven Image Generation

Visual similarity is often measured using traditional metrics such as Structural Similarity Index (SSIM), Peak Signal-to-Noise Ratio (PSNR), and LPIPS [53], which assume that the object is spatially aligned across images. However, in subject-driven generation, this assumption does not hold, as the subject may appear in different poses, positions, and contexts. As a result, it has become common to use global feature-based metrics, such as CLIP-Image [29] and DINO [4], which compare the similarity of image-level embeddings extracted from different models. While these methods are robust to spatial changes, they are inherently global and often fail to capture fine-grained appearance details of the subject. A recent trend involves leveraging Vision-Language Models (VLMs) to evaluate subject-driven generation by prompting them to score specific criteria of the generated images [41, 26]. However, it remains unclear how VLMs form their judgments, and they lack the ability to localize the source of inconsistency within the image. In this work, we aim to bridge this gap by leveraging visual features extracted from the backbones of diffusion models to assess

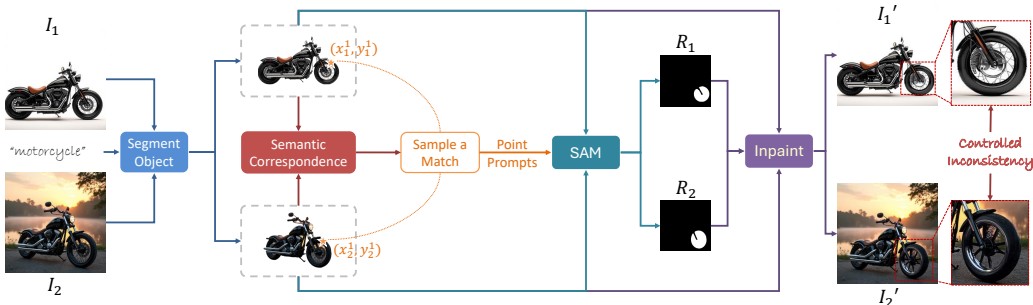

Figure 2: Automated Dataset generation pipeline for producing controlled visual inconsistencies. Access to such pairs of images enables separating visual features from pre-trained backbones in a contrastive manner.

the appearance similarity across images in subject-driven generation. Our approach enables both quantification and spatial localization of visual inconsistencies. By providing a reliable metric that offers fine-grained insight into subject consistency, we take a step toward more robust evaluation and improvement of subject-driven generation.

## 3   Method

In subject-driven image generation, assessing whether different parts of a subject are visually consistent across two images requires matching visual representations that are robust to changes in pose, scale, and environment. This challenge is similar to the semantic correspondence task, where several approaches [39, 22, 51, 52, 42] were trained on SPair-71k [24], which includes annotated semantic points between image pairs of given subjects. As demonstrated in Figure 1, semantic features are insensitive to appearance changes, making them unsuitable for matching visual appearance. To the best of our knowledge, there exists no dataset with annotated visual correspondences between visually similar or dissimilar regions, similar to SPair-71k.

To bridge this gap, we first introduce a novel automated pipeline for constructing a dataset with visual correspondences, leveraging existing datasets of subject-driven image generation such as Subjects200k [41] (Section 3.1). Using the data generated from this pipeline, we then propose an architecture for disentangling semantic and visual representations from the internal features of diffusion models (Section 3.2). Lastly, we introduce a metric that leverages the disentangled features for empirically evaluating visual consistency between image pairs, which quantifies the degree of consistency and localizes inconsistent regions (Section 3.3).

### 3.1   Data Generation Pipeline with Visual Correspondence

We aim to generate a dataset of image pairs with visual correspondences between *visually* similar and dissimilar regions inspired by the SPair-71k dataset. Given a consistent image pair $I_1$ and $I_2$ from a subject-driven dataset, we begin by segmenting the subject in each image using Grounded-SAM [31]. This isolates the subject from the background and makes the computation of correspondences more reliable. Next, we compute semantic correspondences between the two images within the segmented subject regions using the semantic correspondence method CleanDIFT [39]. Specifically, we extract features from the sixth decoder layer of the diffusion model UNet $\Phi$, which has been shown to contain semantically rich information [42, 39], resulting in $F_1^6 = \Phi(I_1)$ and $F_2^6 = \Phi(I_2)$. We then compute the pairwise similarity between the two feature maps to form a similarity matrix $\mathcal{D} = F_1^6 F_2^{6^T}$. Corresponding points are obtained by selecting locations with the highest similarity scores using $\arg\max \mathcal{D}$, to obtain the following correspondence mapping:

$$C_1 = \langle (x_1^1, y_1^1), \ldots, (x_1^N, y_1^N) \rangle \ , \qquad C_2 = \langle (x_2^1, y_2^1), \ldots, (x_2^N, y_2^N) \rangle \ , \qquad C_1, C_2 \in \mathrm{R}^{N \times 2} \quad (1)$$

where $N$ denotes the number of correspondences.

Based on these correspondences, we aim to match semantically similar regions across both images and then alter them visually to generate a pair of images with known, localized visual inconsistencies. To

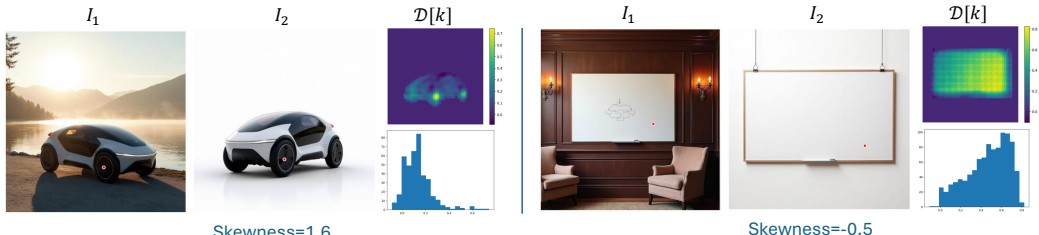

Figure 3: Examples illustrating how the skewness of the matching scores distribution correlates with matching ambiguity. High skewness implies a distinct match, while low skewness indicates diffuse or ambiguous correspondences.

achieve this, we sample a point $k$ from $C_1$ that has a high similarity score and use the corresponding point pair $(x_1^k, y_1^k)$ and $(x_2^k, y_2^k)$ as prompts to the SAM model [16], which segments a localized region in each image. This produces region masks defined as $R_i = \text{SAM}(I_i, (x_i^k, y_i^k))$. We configure SAM to return multiple candidate masks and select the one with the smallest area, as it is more likely to correspond to an isolated semantic part rather than the entire object.

**Handling Ambiguous Matches:** When segmenting regions, a common challenge arises with flat-textured subjects, such as a whiteboard, where the selected point $k$ tends to match broadly across the object (see Figure 3), leading SAM to segment the entire subject rather than a localized part. To address this, we propose using the skewness of the similarity distribution $\mathcal{D}[k]$ to identify ambiguous matches. The skewness is computed as:

$$\text{Skewness}(\mathcal{D}[k]) = \frac{n}{(n-1)(n-2)} \sum_{i=1}^{n} \left( \frac{\mathcal{D}[k] - \mu}{\sigma} \right)^3 \tag{2}$$

where $n = |\mathcal{D}[k]|$, $\mu = \text{mean}(\mathcal{D}[k])$, and $\sigma = \text{STD}(\mathcal{D}[k])$. We observe that high skewness corresponds to matches in textured regions, characterized by a long-tailed distribution and low ambiguity. In contrast, low skewness indicates more uniform similarity scores, suggesting ambiguous matches typically associated with flat surfaces as illustrated in Figure 3.

**Validating Matched Regions:** We apply additional heuristics to alleviate segmentation failures and to ensure that the selected regions in both images correspond to the same semantic part, including constraints on aspect ratio and relative size with respect to the full object. For regions that pass all checks, we perform inpainting using a diffusion-based inpainting model (we use SDXL [27] for efficiency), to obtain two inconsistent images $I_1'$ and $I_2'$. Specifically, we crop a patch around the region with padding and feed it to the inpainting model to ensure that the model does not observe the rest of the object, promoting the generation of visually distinct content. To confirm that the inpainted region differs from the original, we compute the LPIPS score [53] between the original and inpainted regions in $I_i$ and $I_i'$. We discard samples with low LPIPS scores to ensure meaningful variation. Further details on the filtering strategy are provided in the supplementary material. We provide an illustration for the data generation pipeline in Figure 2. Eventually, each dataset sample comprises a consistent image pair $(I_1, I_2)$, an inconsistent pair $(I_1', I_2')$ generated with our data generation pipeline, subject masks $(O_1, O_2)$, inpainted region masks $(R_1, R_2)$, subject prompt $s$ and target prompt $p$.

### 3.2 Learning Disentangled Semantic and Visual Representations

Given an inconsistent image pair $I_1'$ and $I_2'$, which were generated with our automated pipeline in Section 3.1, where the inconsistent regions are defined by the masks $R_1$ and $R_2$, we aim to disentangle semantic and visual features of the diffusion model backbone. We start by extracting multi-layer features for both images using the diffusion model backbone $\Phi$, yielding feature maps $F_1 = \Phi(I_1')$ and $F_2 = \Phi(I_2')$ that include features from multiple decoder layers $l \in L$. To aggregate features from the different layers, we follow the approach of [22], but instead of using a single aggregation network, we employ two separate networks, $\Psi_s^l$ and $\Psi_v^l$, to aggregate semantic and visual representations separately. Each aggregation network encompasses a ResNet [11] block per decoder layer that are

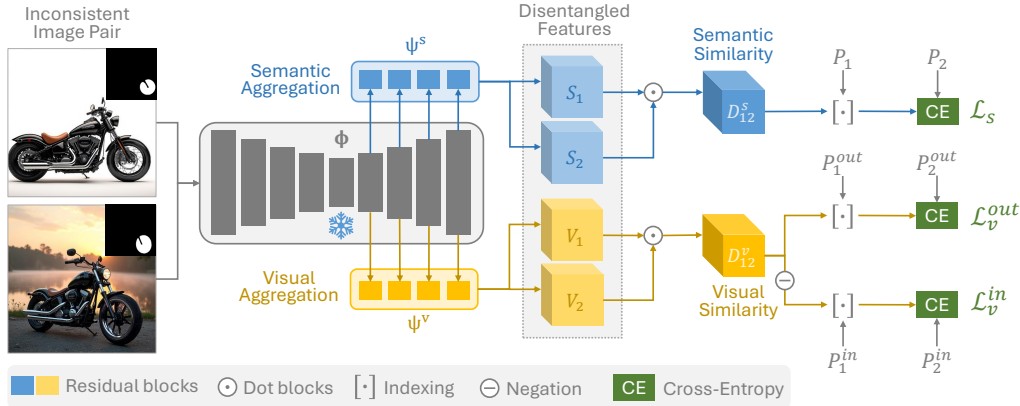

Figure 4: Overview of the proposed architecture for disentangling semantic and visual features from a frozen diffusion backbone $\Phi$. Inconsistent regions between the input image pairs are indicated by binary masks (left). The semantic branch (blue) encourages the features of corresponding semantic points in both images, $P_1$ and $P_2$, to align. In the visual branch (yellow), we bring visually consistent points *outside* the inpainted regions, $P_1^{\text{out}}$ and $P_2^{\text{out}}$, closer together, while pushing apart features at inconsistent points *inside* the inpainted regions, $P_1^{\text{in}}$ and $P_2^{\text{in}}$.

combined using trainable scalar weights $w^l$:

$$S_i = \sum_l^L w_s^l \ \Psi_s^l(F_i^l) \ , \qquad V_i = \sum_l^L w_v^l \ \Psi_v^l(F_i^l) \ , \qquad S_i, V_i \in \mathrm{R}^{d \times q} \ , \tag{3}$$

where $S_i$, $V_i$ are the semantic and visual features respectively, $i \in \{1, 2\}$, $d = w \times h$ denotes the flattened spatial dimensions, and $q$ is the feature dimensionality. This architecture allows for flexible selection of features that capture either semantic content or visual appearance. An overview of the architecture is provided in Figure 4.

Next, we compute cyclic similarity matrices between the aggregated semantic and visual features of the two images using dot products:

$$\mathcal{D}_{12}^s = S_1 S_2^T \ , \qquad \mathcal{D}_{21}^s = S_2 S_1^T \ , \qquad \mathcal{D}_{12}^s, \mathcal{D}_{21}^s \in \mathrm{R}^{d \times d} \ , \tag{4}$$

$$\mathcal{D}_{12}^v = V_1 V_2^T \ , \qquad \mathcal{D}_{21}^v = V_2 V_1^T \ , \qquad \mathcal{D}_{12}^v, \mathcal{D}_{21}^v \in \mathrm{R}^{d \times d} \ , \tag{5}$$

Our objective is to encourage semantic features to be similar across all point correspondences, while visual features should be similar only outside the inconsistent regions $R_i$ and dissimilar within them. To achieve this, we adopt a contrastive learning framework that brings similar features closer and pushes dissimilar ones apart. As a first step, we partition the set of pre-computed correspondences $C_i$ from Section 3.1 into two subsets: points that fall inside and outside the inconsistent regions:

$$P_i^{\text{in}} = \left\{ (x_i^j, y_i^j) \ \middle| \ (x_i^j, y_i^j) \in C_i, \ R_i(x_i^j, y_i^j) = 1 \right\} \ , \tag{6}$$

$$P_i^{\text{out}} = \left\{ (x_i^j, y_i^j) \ \middle| \ (x_i^j, y_i^j) \in C_i, \ R_i(x_i^j, y_i^j) = 0 \right\} \ . \tag{7}$$

Then, inspired by the contrastive objective used in CLIP [29], we define a semantic correspondence loss as follows:

$$\mathcal{L}_s = \texttt{CrossEntropy}\big(\mathcal{D}_{12}^s(P_1), \ P_2\big) \ , \quad P_i = P_i^{\text{in}} \cup P_i^{\text{out}} \ . \tag{8}$$

This semantic loss encourages matched points, whether inside or outside the inconsistent region, to exhibit similar semantic feature representations.

To disentangle appearance-specific features, we define a visual loss that explicitly separates consistent from inconsistent regions. For inconsistent regions, we use a negative similarity objective:

$$\mathcal{L}_v^{\text{in}} = \texttt{CrossEntropy}\big(-\mathcal{D}_{12}^v(P_1^{\text{in}}), \ P_2^{\text{in}}\big) \ . \tag{9}$$

This term penalizes similarity in appearance features for points known to be visually inconsistent. For consistent regions, we retain the standard contrastive loss to encourage matching:

$$\mathcal{L}_v^{\text{out}} = \text{CrossEntropy}\big(\mathcal{D}_{12}^v(P_1^{\text{out}}),\ P_2^{\text{out}}\big)\ . \tag{10}$$

Additionally, visually corresponding points inside the inpainted regions of the original consistent pairs $I_1$ and $I_2$ can be incorporated into Equation (10) to provide additional supervision; however, we omit the formal definition of this part for simplicity. All loss terms are also computed in the reverse direction as well using $\mathcal{D}_{21}^s$ and $\mathcal{D}_{21}^v$, and each loss term is averaged over both directions. The final training objective combines semantic and visual losses as follows:

$$\mathcal{L} = \mathcal{L}_s + \alpha(\mathcal{L}_v^{\text{in}} + \mathcal{L}_v^{\text{out}}) \tag{11}$$

where $\alpha$ is a scaling factor used to prioritize the visual branch, since the semantic branch can already be extracted reliably even without any aggregation, as shown in [42, 39].

### 3.3 A Metric for Evaluating Subject-Driven Image Generation

Having disentangled visual and semantic features from the backbone of pre-trained diffusion models in the previous section, we now aim to leverage these features to estimate and localize visual consistency in subject-driven image generation. Given two test images $I_1$ and $I_2$ generated by a subject-driven generation method, our goal is to evaluate the visual consistency of the subject across the two images.

We begin by passing both images through our architecture to extract semantic and visual feature maps, denoted as $F_i^s$ and $F_i^v$, respectively, following the aggregation described in Equation (3). As in the previous section, we compute pairwise similarities between features to obtain semantic and visual similarity matrices, denoted by $\mathcal{D}^s$ and $\mathcal{D}^v$, respectively. We then take the maximum similarity score for each point to obtain $\hat{\mathcal{D}}^s = \max(\mathcal{D}^s)$ and $\hat{\mathcal{D}}^v = \max(\mathcal{D}^v)$, representing the best per-point match in the semantic and visual domains.

Semantic correspondences are identified by selecting points whose semantic similarity exceeds a predefined threshold $\mathcal{T}_s$, *i.e.*, $\hat{\mathcal{D}}^s > \mathcal{T}_s$, resulting in a set of confident semantic matches indexed by $\mathcal{J}_s$. This filtering ensures that we only consider regions that are both visible and semantically coherent. We set $\mathcal{T}_s = 0.7$ in all experiments.

To assess visual consistency at these semantically matched locations, we use the same index set $\mathcal{J}_s$ to retrieve the corresponding visual similarity scores from $\hat{\mathcal{D}}^v$. Given a visual similarity threshold $\mathcal{T}_v$, we define the *Visual Semantic Match (VSM)* metric as:

$$\text{VSM}(\mathcal{T}_v) = \frac{1}{|\mathcal{J}_s|} \sum_{j \in \mathcal{J}_s} \delta\left[\hat{\mathcal{D}}_j^v > \mathcal{T}_v\right] \tag{12}$$

where $\delta[\cdot]$ is the indicator function, equal to 1 if the condition holds and 0 otherwise.

This metric captures the proportion of semantically aligned regions that are also visually consistent, providing a quantitative measure of consistency between the two generated images. Inconsistent regions can be identified by subtracting visually matched locations from the semantically matched ones, or by examining regions in $\hat{\mathcal{D}}^v$ with low visual similarity scores.

## 4 Experiments

In this section, we evaluate the effectiveness of our approach and compare it against commonly used feature-based metrics such as CLIP [29] and DINO [4], as well as VLM-based approaches (*e.g.* ChatGPT-4o). Then, we provide an ablation study to analyze the impact of key architectural design choices and provide further insights into the behavior of our model.

### 4.1 Implementation Details

**Dataset:** We use the data generation pipeline described in Section 3.1 to construct the dataset used to train our architecture. The pipeline takes consistent image pairs from the Subjects200k dataset [41] as input, although any subject-driven generation dataset can be used.

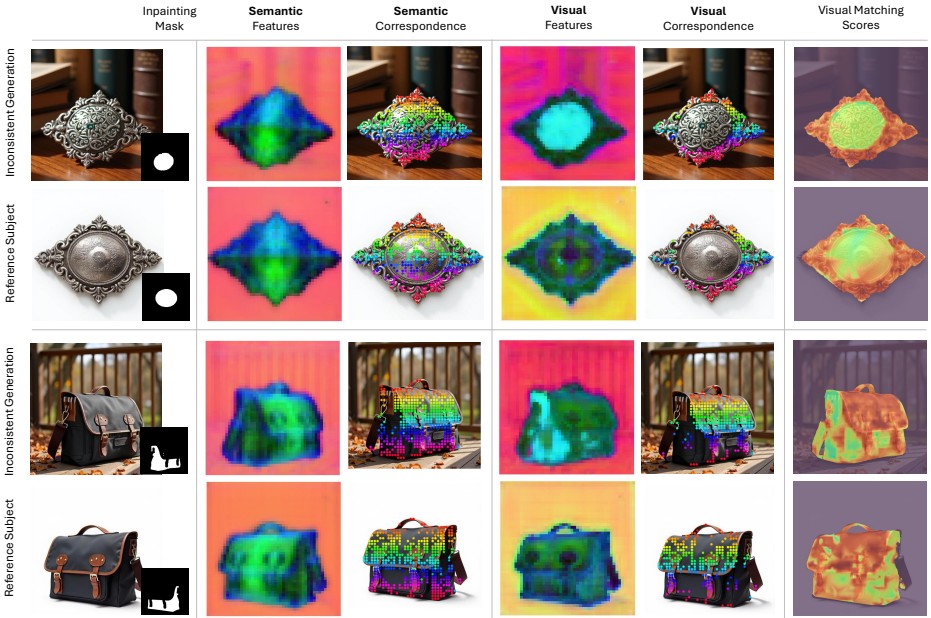

Figure 5: Qualitative examples of semantic and visual features, along with their correspondences, produced by our architecture. Regions that fall within the inpainting mask exhibit visually dissimilar features, enabling the detection of visual inconsistencies based on feature similarity. Dark Red is most consistent and Yellow is least consistent.

During generation, we apply several filtering steps to ensure high-quality training data. We discard samples with matching skewness below 1.3 (see Figure 3), and we enforce that the inconsistent region occupies between 5% and 60% of the subject area. Additionally, we filter out samples where the inpainted region has an LPIPS score below $0.15$ to ensure sufficient visual inconsistency. The final dataset consists of 5,000 image pairs for training and 500 image pairs for validation.

*Testset:* We generate a testset of 100 samples using our dataset, and we manually revise it to ensure the reliability of evaluation.

**Hyperparameters:** We use the backbone of Stable Diffusion 2.1 [32] for all experiments following CleanDIFT [39]. We set the spatial resolution of the aggregated features in Equation (3) to $h = w = 48$, resulting in $d = 48 \times 48 = 2304$, following CleanDIFT [39]. The projected feature dimensionality is set to $q = 384$, as in [22]. We empirically found that setting $\alpha = 10$ in Equation (11) yields the best performance. We train the model for 30 epochs using the AdamW optimizer [21] with a learning rate of 1e-3 that is divided by 10 every 10 epochs. The training is done on 1 A100 GPU (40GB) and takes 12 hours. The source code for the dataset generation pipeline and the proposed architecture are publicly available. [1]

### 4.2 Evaluating the VSM Metric

To validate our proposed *Visual-Semantic Match (VSM)* metric, described in Section 3.3, we first perform a controlled evaluation where we define an oracle based on the inpainted regions for each image pair in the test set. This oracle is defined based on the ratio of the inconsistent region $R_1$ to the total object mask $O_1$ (see Figure 2), computed as:

$$\text{Oracle} = 1 - \left( \frac{\sum R_1}{\sum O_1} \right) \tag{13}$$

This oracle reflects the ground-truth degree of visual consistency between the image pair. An effective metric should produce estimates that strongly correlate with the oracle. We evaluate against commonly

---

[1]https://github.com/abdo-eldesokey/mind-the-glitch

|          | *Controlled Inconsistency* | | | | *Subject-Driven Generation* | | | |
|----------|-------|-------|-------|------------|-------|-------|-------|------------|
|          | CLIP  | DINO  | VLM*  | VSM (Ours) | CLIP  | DINO  | VLM*  | VSM (Ours) |
| Pearson  | -0.053 | 0.087 | 0.072 | **0.448**  | 0.156 | 0.164 | 0.079 | **0.405**  |
| Spearman | -0.005 | 0.120 | 0.091 | **0.582**  | 0.112 | 0.146 | 0.073 | **0.369**  |

Table 1: Average correlation scores across methods. Our VSM achieves significantly higher correlation with the oracle than other metrics both on controlled and realistic settings. *(ChatGPT-4o)

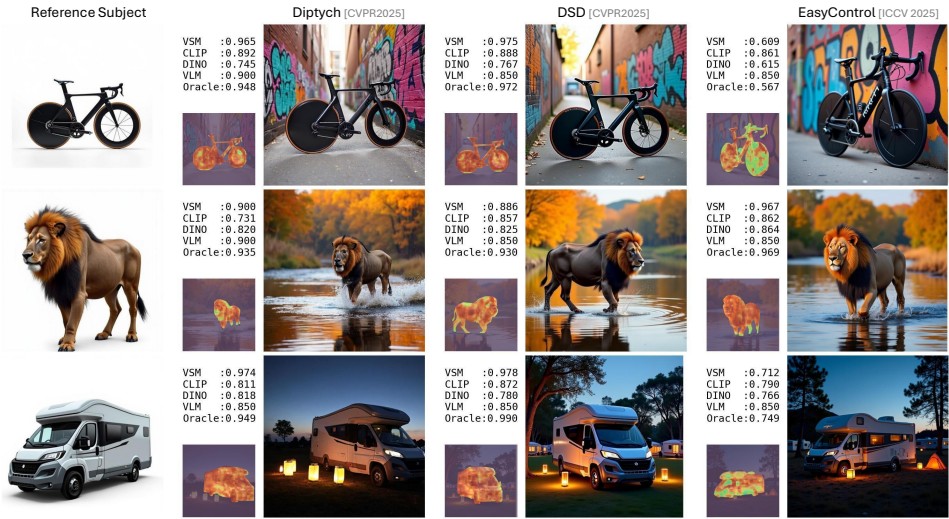

Figure 6: Qualitative examples of evaluating subject-driven image generation approaches using our proposed VSM metric and other existing approaches. Our VSM metric can accurately quantify and localize inconsistency and is more consistent with the oracle. Dark Red is most consistent and Yellow is least consistent.

used metrics in the literature for comparing image pairs: CLIP and DINO image-to-image similarity, as well as ChatGPT-4o as a representative Vision-Language Model (VLM). For VLM, we adopt the prompt used in [26], but modify it to produce a numerical score in the range of 0 to 100.

We report Pearson and Spearman correlations between each evaluated metric and the oracle in Table 1 (left). Our VSM metric shows significantly higher correlation compared to both feature-based similarity metrics and VLM judgments, indicating its greater reliability for measuring visual consistency. Qualitative results in Figure 5 illustrate semantic and visual features with their correspondences. While semantic features align with semantically similar regions regardless of appearance, visual features differ notably within inpainted areas and do not match across image pairs. This is evident in the score heatmaps shown in the rightmost column.

### 4.3 Evaluating Subject-Driven Image Generation

To evaluate the effectiveness of our VSM metric in a *realistic* subject-driven image generation setting, we evaluate three recent methods: Diptych [38], DSD-Diffusion [3], and EasyControl [54]. Using the reference image $I_1$ and target prompt $p$ from our test set, we generate 100 images per method and evaluate them using VSM and other metrics as before. To compute the oracle, we manually annotate the generated images to mark visually inconsistent regions and calculate the oracle score as defined in Equation (13).

Table 1 shows that our VSM metric consistently outperforms all other metrics, demonstrating strong generalization to real-world subject-driven generation. We also present qualitative examples in Figure 6, illustrating that VSM aligns most closely with the oracle. This is further supported by the KDE plots in Figure 7, where VSM shows the highest agreement with the oracle, while VLM tends to score most image pairs between 75–95 regardless of visual discrepancies.

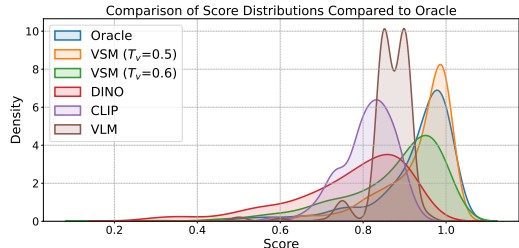

Figure 7: Score Distribution of different metrics compared to the Oracle.

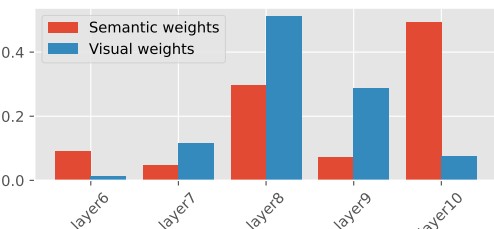

Figure 8: Aggregation weights.

|  | $\mathcal{T}_v = 0.5$ | $\mathcal{T}_v = 0.7$ | $\alpha = 1$ | Skewness$> 1.0$ | Skewness$> 1.5$ | VSM (Ours) |
|---|---|---|---|---|---|---|
| Pearson | 0.465 | 0.352 | 0.118 | 0.232 | 0.224 | **0.448** |
| Spearman | 0.454 | 0.496 | 0.104 | 0.250 | 0.225 | **0.582** |

Table 2: Ablation analysis of different hyperparameters.

## 4.4 Ablation Analysis

We visualize the learned aggregation weights for the semantic and visual branches in Figure 8. Visual features are primarily derived from decoder layers 8 and 9, while semantic features draw from layers 8 and 10, indicating that certain layers (e.g., layer 8) contribute to both representations. Notably, our learned semantic weights differ from those in [22], likely due to differences in supervision: their model is trained on sparse keypoints, whereas ours uses dense correspondences, favoring later layers with higher spatial resolution.

Table 2 presents an ablation study of key hyperparameters. For the VSM similarity threshold $\mathcal{T}_v$, both low (0.5) and high (0.7) values reduce correlation with the oracle, with $\mathcal{T}_v = 0.6$ yielding the best performance. Setting $\alpha = 1$ in the training loss significantly degrades performance by reducing emphasis on visual features in favor of semantic features, which are easier to learn and already present in diffusion backbones without additional training [42, 39]. In the dataset pipeline, high skewness thresholds overly restrict sample diversity, while low thresholds allow ambiguous matches—both leading to performance drops.

## 5 Limitations and Future Work

Our feature disentanglement is inherently partial, as visual features may still carry semantic information. Achieving full disentanglement, enabling visual matching across semantically different objects, remains a promising future direction. The quality of the learned features also depends on the reference dataset used in the automated pipeline. The Subjects200k dataset [41], while large-scale, was validated automatically and may include noisy pairs. Manual curation or improved filtering could enhance supervision quality. Additionally, detecting fine-grained inconsistencies is further limited by the spatial resolution of the diffusion features. We plan to address this via multi-scale aggregation and higher-resolution extraction. Lastly, our current framework targets structural and appearance-level inconsistencies. Extending it to handle broader variations, such as artistic style or color, may require further decomposing. We include failure cases in the supplementary to guide future work.

## 6 Conclusion

We proposed a framework for disentangling diffusion model features into semantic and visual components. Using an automated dataset pipeline with controlled visual inconsistencies, we trained a contrastive architecture to separate the two. This enables visual correspondence across images, complementing semantic matching, and supports fine-grained analysis. Leveraging this, we introduced a metric that quantifies and localizes inconsistencies in subject-driven generation. Our approach outperforms existing metrics, offering a more reliable and interpretable evaluation framework.

## Acknowledgment

The research reported in this publication was supported by funding from King Abdullah University of Science and Technology (KAUST) - Center of Excellence for Generative AI, under award number 5940.

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
