# OpenReview forum: "Mind-the-Glitch: Visual Correspondence for Detecting Inconsistencies in Subject-Driven Generation"
_NeurIPS.cc/2025/Conference — NeurIPS 2025 spotlight_

### Official Review · Reviewer_SfLC · 2025-06-10

**Clarity:** 3
**Significance:** 2
**Originality:** 3
**Rating:** 4
**Confidence:** 4

**Summary:**

The paper proposes learning semantic and visual features from diffusion features. This is done by tuning diffusion features via contrastive loss on generated data. The data generation pipeline consists of taking a (visually and semantically) consistent pair of images, and applying segmentation (to detect consistent regions between the images) and inpainting (to alter one of the consistent regions), and as a result constructing an inconsistent pair.

**Questions:**

Please see weaknesses.

**Ethical Concerns:**

["NO or VERY MINOR ethics concerns only"]

**Final Justification:**

Thanks to the authors for the response.
While I still have some concerns about the sensitivity of the features (e.g., in the last two rows of figure 3, when the pose changes significantly, VSM score is much lower), the authors have addressed most of my concerns.

The additional experiments and user study show that the features learn beyond the inpainted regions, and the paper provides new dataset and metric which can benefit the community.

Therefore, I am happy to increase my score. I highly suggest the authors to add an analysis about the sensitivity of VSM to pose, illumination, scale and image degradation, that would encourage other researchers to adopt the metric.

**Limitations:**

Please see weaknesses.

**Paper Formatting Concerns:**

No concerns.

**Quality:**

2

**Strengths And Weaknesses:**

**Strengths**

1.	The paper is clearly written.
2.	The method and its motivation is well explained.
3.	The data generation pipeline is interesting and novel, enabling the creating of inconsistent pairs of images for training.
4.	Detecting visual inconsistent regions is beneficial for subject-based image generation, and visual and semantic features can possibly be used to enhance the effectiveness of such applications.

**Weaknesses**

1.	Unclear generalization abilities of the learned features: my major concern is that it is unclear whether the tuned features truly learn to detect visual inconsistencies, or rather to identify inpainted regions (and ignore them/assign some random features to them so there won’t be correspondences between the inconsistent pair in these regions). While the inpainted images may appear natural to the human eye (although I can see blurry and unrealistic examples of inpainted regions), a strong neural net can probably easily detect stitched inpainted regions and treat them differently.
* The evaluation of the method relies on the same data generation pipeline, and thus it is unclear whether the learned features actually generalize and learn visual features. I believe it is crucial to create a creative test set of real, unaltered inconsistent pairs for evaluation. Examples for such pairs can be different images of a person with different clothing, images of a house with different decorations (e.g., holiday decorations), etc.
* Another valuable experiment is to collect pairs of images, created by the data generated pipeline, where the inpainted region is very close (visually) to the original image. In this case, we would still expect to see visual correspondences between the “inconsistent pair”.
* In some examples, it seems that there are no correspondences in the inpainted region between the images, even when some parts within the inpainted region still look visually similar between the images (e.g., the rings in the main paper).
* Additionally, it will be helpful if the authors can also visualize the inpainting mask used to create the inconsistent pair in figure 6, for reference.
2.	Lacking experiments on real tasks or benchmarks: The main motivation presented by the authors for detecting inconsistent regions is subject-driven image generation. However, another concern regarding the evaluation section is that the paper lacks experiments on a real task. It would be significantly more convincing if the authors evaluated the performance of their features on subject-driven generation methods or personalized-generation methods (e.g., textual inversion [1], dreambooth [2], RealFill [3]), aiming to generate images based on a set of reference images, and demonstrated the effectiveness of their visual and semantic features in actually detecting inconsistent regions.

3. Limitations: The authors discuss limitations regarding the training data and inconsistencies between fine-details which the features may not detect. I think it would be helpful if the authors can include failure cases, and an analysis regarding the sensitivity of the learned features to real-world image degradations as motion blur, noise, photometric distortions, illumination changes.

**References**

[1] An Image is Worth One Word: Personalizing Text-to-Image Generation using Textual Inversion, ICLR 2023.

[2] DreamBooth: Fine Tuning Text-to-Image Diffusion Models for Subject-Driven Generation, CVPR 2023.

[3] RealFill: Reference-Driven Generation for Authentic Image Completion, SIGGRAPH 2024

---

> ### Author Rebuttal · Authors · 2025-07-29
>
> &emsp; We thank the reviewer for the thorough feedback. We appreciate the recognition of our **paper’s clarity**, **the novelty of the data generation pipeline**, and **the relevance of our approach to subject-driven image generation**.
>
> We address the reviewer’s concerns and questions below.
>
> ---
>
> > **“Unclear generalization abilities of the learned features: my major concern is that it is unclear whether the tuned features truly learn to detect visual inconsistencies, or rather to identify inpainted regions ...”**
>
> &emsp; We fully understand and appreciate this concern; it is one we explicitly considered in the design of our method.
>
> First, to minimize the risk of the network learning superficial cues such as **boundary artifacts from inpainting**, we carefully designed our inpainting pipeline to operate on **cropped patches with padding** rather than applying inpainting to the full image. This design significantly reduces visible seams or texture discontinuities at the boundaries. Secondly, since our objective is **point-based rather than patch-based**, most training points lie within the central region of the inpainted patch, away from any potential boundary artifacts.
>
> Nevertheless, as pointed out by the reviewer, the networks may still learn to exploit subtle, invisible cues undetectable to human observers. To invalidate this hypothesis, we evaluated our network on **real subject-driven generation outputs** in **Section D and Figure 3 of the supplementary material**. These images are produced by three different subject-driven generation methods, EasyControl, DSD Diffusion, and Diptych Prompting, **without any synthetic perturbations or inpainting**.
>
> In these settings, Figure 3 in the supplementary shows that our method **successfully localizes inconsistent regions**, whereas existing metrics often fail to distinguish between models. This demonstrates that our learned features are not overfitting to inpainting artifacts and instead **generalize effectively to natural inconsistencies** in real subject-driven generation scenarios.
>
> ---
>
> > **“The evaluation of the method relies on the same data generation pipeline, and thus it is unclear whether the learned features actually generalize and learn visual features. I believe it is crucial to create a creative test set of real, unaltered inconsistent pairs for evaluation ...”**
>
> &emsp; We agree that evaluating on real, unaltered, inconsistent image pairs is essential for assessing generalization beyond the synthetic training pipeline. As mentioned above, we include such an evaluation in **Section D and Figure 3 of the supplementary material**, where we apply our method to outputs from multiple **real subject-driven generation models**.
>
> These examples are **not generated through our inpainting pipeline**, but instead reflect natural inconsistencies such as changes in pose, texture, and object details. As shown in Figure 3 in the supplementary, our method **successfully localizes inconsistent regions** in these real-world cases, whereas existing metrics such as CLIP, DINO, and VLM often assign similar or overly optimistic scores across models.
>
> We believe this evaluation provides strong evidence that our learned features **generalize beyond synthetic artifacts** and are effective at capturing meaningful visual inconsistencies in real subject-driven generation scenarios.
>
> ---
>
> > **- “In some examples, it seems that there are no correspondences in the inpainted region between the images, even when some parts within the inpainted region still look visually similar between the images...”**
> > **- “Another valuable experiment is to collect pairs of images, created by the data generated pipeline, where the inpainted region is very close (visually) to the original image. In this case, we would still expect to see visual correspondences between the ‘inconsistent pair’.”**
>
> These are excellent observations, and both behaviors are primarily governed by two key mechanisms in our pipeline:
>
> 1. **Visual similarity threshold ($\tau$)**:
>    As described in Section 3.3 (line 232), when computing visual correspondences, we apply a similarity score threshold $\tau$ to determine which matches are retained. A higher threshold results in stricter matching; even minor visual discrepancies may prevent a correspondence, while a lower threshold is more forgiving and allows matches between regions that are only approximately similar in appearance.
>
>    In our experiments, we found $\tau = 0.6$ to strike a good trade-off between sensitivity and robustness, as supported by the ablation study in Section 4.3. However, this parameter is fully tunable and could be adapted depending on the application’s sensitivity requirements. A natural extension of our metric would be to report **multiple VSM scores at different $\tau$ values**, such as $VSM_{\tau=0.5}$, $VSM_{\tau=0.6}$, and $VSM_{\tau=0.7}$, to give users a more comprehensive picture of subject-driven generation performance across varying thresholds.
>
> 2. **LPIPS-based filtering in training data**:
>    To ensure that the inpainted regions used for training reflect meaningful changes, we compute the LPIPS distance between each inpainted patch and its corresponding original patch. Pairs with LPIPS scores below a set threshold are discarded. This ensures that the model focuses on learning **clear, perceptually relevant inconsistencies**, rather than becoming overly sensitive to minimal or imperceptible changes.
>
>    Based on the reviewer’s suggestion, we experimented with different LPIPS thresholds and report below their effect on correlation with the oracle as explained in Section 4.3 in the main paper:
>
>    | LPIPS Threshold | Pearson Correlation | Spearman Correlation |
>    |----------------|---------------------|-----------------------|
>    | 0.10           | 0.3523              | 0.3935                |
>    | 0.15           | **0.4750**          | **0.4410**            |
>    | 0.20           | 0.2330              | 0.1790                |
>
>    These results show that a **threshold of 0.10** admits inpainted regions that are too similar to the original, weakening the model's ability to learn discriminative features. Conversely, a **threshold of 0.20** removes too many moderately different regions, limiting variability. We selected **0.15** as it offered the best balance between preserving useful variation and excluding near-duplicates, yielding the highest correlation with the oracle.
>
> Both $\tau$ and the LPIPS threshold are **exposed and tunable**, offering users flexibility to tailor the system to their desired level of tolerance or strictness.
>
> ---
>
> > **“Additionally, it will be helpful if the authors can also visualize the inpainting mask used to create the inconsistent pair in figure 6, for reference.”**
>
> That is a good suggestion and would definitely help interpret the figure better. We will incorporate this suggestion in the revised version of the paper.
>
> ---
>
> > **“Lacking experiments on real tasks or benchmarks: The main motivation presented by the authors for detecting inconsistent regions is subject-driven image generation. However, another concern regarding the evaluation section is that the paper lacks experiments on a real task. It would be significantly more convincing if the authors evaluated the performance of their features on subject-driven generation methods or personalized-generation methods…”**
>
> &emsp; We appreciate this suggestion and agree that evaluation on real subject-driven generation outputs is critical for validating the utility of our method. This experiment **was provided in Section D of the supplementary material** (“Benchmarking Subject-Driven Generation Approaches with Our Metric”), where we apply our approach to outputs from three recent subject-driven generation methods: **Diptych Prompting**, **DSD Diffusion**, and **EasyControl**.
>
> These models generate subject-specific images based on reference inputs or identity prompts, without any artificial perturbations or inpainting. As shown in **Figure 3 (supplementary)**, our evaluation reveals that existing metrics, including CLIP, DINO, and VLM (ChatGPT-4o), often assign overly optimistic or indistinguishable scores across methods, despite clearly visible inconsistencies. In contrast, our **VSM metric provides more discriminative and spatially grounded scores**, aligning more closely with human perception of visual consistency.
>
> Due to space limits, we also refer the reviewer to our answer to **Question #2 of Reviewer bVqH** that shows that our metric achieves the highest correlation with human judgment in real subject-driven generation.
>
> ---
>
> > **“Limitations: The authors discuss limitations regarding the training data and inconsistencies between fine-details which the features may not detect. I think it would be helpful if the authors can include failure cases, and an analysis regarding the sensitivity of the learned features to real-world image degradations as motion blur, noise, photometric distortions, illumination changes.”**
>
> &emsp; We thank the reviewer for this suggestion. Due to the recent NeurIPS rebuttal policy changes, we are unable to include new visual examples or failure case analyses at this stage. However, we fully agree that examining the sensitivity of our learned features to real-world image degradations, such as motion blur, noise, and illumination changes, is an important direction.
>
> In the revised version of the paper, we will include **qualitative failure cases** as well as an initial analysis of how the method responds to various photometric and geometric distortions. This will help characterize the **robustness and limitations** of the learned visual features in less controlled settings, and guide future extensions of the approach toward more degradation-resilient representations.

---

> ### Comment · Reviewer_SfLC · 2025-08-01
>
> Thank you to the authors for the detailed response.
>
> Thank you for the clarification about additional experiments in the supplementary material. I indeed missed those, as I did not see a reference to them in the main paper. I suggest adding at least a small section about these importand experiments in the main paper as they could encourage other researchers to adopt VSM as a metric.
>
> These experiments, along with your responses to my questions and to Q2 of reviewer bVqH, support the claim that the features learn visual properties beyond recognizing the inpainted regions and address my major concerns.
>
> I can see cases where the sensitivity of the features is unclear. For example, in the first row of figure 3 in the supplementary material, it seems that the inconsistencies between the reference image and the leftmost generated image are less drastic than the rightmost generated image. Yet, VSM score indicate the opposite, perhaps due to a different pose of the front wheel.
>
> Investigating the sensitivity of the features to changes in pose, scale, illumination, noise and blur is important for general applicability and I highly encourage the authors to include such analysis in the revised version (along with failure cases). Other qualitative examples, in the same format of figure 3, would also be valuable. Yet, I understand this is not possible in the current rebuttal and do no expect that.
>
> Before finalizing my score, I would like to ask about your response to Q2 of reviewer bVqH: Which models were used for subject-driven generation in the user study? Did you include all three baselines from section D? What is the alignment between VSM ranking of the subject-driven methods and the human preferences?

---

> > ### Author Response · Authors · 2025-08-04
> >
> > We thank the reviewer for the response.
> >
> > > **“Thank you for the clarification about additional experiments in the supplementary material. I indeed missed those, as I did not see a reference to them in the main paper. I suggest adding at least a small section about these importand experiments in the main paper as they could encourage other researchers to adopt VSM as a metric.”**
> >
> > We missed referencing the benchmark results in the supplementary material within the main paper, and we appreciate the reviewer bringing this to our attention.
> >
> > We will **move the benchmark results to the main paper** in the revised version to better demonstrate the **generalizability of the VSM metric** and encourage broader adoption of our metric.
> >
> > ---
> >
> > > **“These experiments, along with your responses to my questions and to Q2 of reviewer bVqH, support the claim that the features learn visual properties beyond recognizing the inpainted regions and address my major concerns.”**
> >
> > We are glad that the benchmark experiments and the user study, as requested by reviewer bVqH, have addressed your concerns regarding the **generalization of our approach to real subject-driven generation scenarios**.
> >
> > ---
> >
> > > **“I can see cases where the sensitivity of the features is unclear. For example, in the first row of figure 3 in the supplementary material, it seems that the inconsistencies between the reference image and the leftmost generated image are less drastic than the rightmost generated image. Yet, VSM score indicate the opposite, perhaps due to a different pose of the front wheel.”**
> >
> > In the example referenced, the VSM score for the **leftmost image (Diptych)** is **0.698**, which is higher than the score for the **rightmost image (EasyControl)** at **0.607**. This reflects that, overall, Diptych achieves better consistency than EasyControl.
> >
> > As noted by the reviewer, Diptych is penalized for the **drastic change in pose of the front wheel**, while EasyControl is penalized because it **generates a single front wheel** instead of two.
> >
> > It is important to note that the **inconsistency heatmaps are self-normalized per image**, which means that the most inconsistent regions are highlighted relative to other regions within the same image. However, the VSM score aggregates all inconsistencies and allows comparison across images.
> >
> > ---
> >
> > > **“Investigating the sensitivity of the features to changes in pose, scale, illumination, noise and blur is important for general applicability and I highly encourage the authors to include such analysis in the revised version (along with failure cases)...”**
> >
> > We thank the reviewer again for this suggestion. We fully agree that analyzing the **sensitivity of the learned features** to variations in pose, scale, illumination, noise, and blur would provide valuable insights into their robustness and general applicability.
> >
> > In the **revised version of the paper**, we will include a new figure following the same format as **Figure 3**, evaluating how these real-world variations affect the **VSM metric**. We will also include **qualitative failure cases** to highlight the limitations and guide future improvements.
> >
> > ---
> >
> > > **“Before finalizing my score, I would like to ask about your response to Q2 of reviewer bVqH: Which models were used for subject-driven generation in the user study? Did you include all three baselines from section D? What is the alignment between VSM ranking of the subject-driven methods and the human preferences?**
> >
> > For the user study, due to the time-consuming and relatively high cost of manual annotation, we selected **Diptych** as a representative **training-free approach** and **EasyControl** as a representative **trained approach**. These two methods capture diversity in design while keeping the annotation workload feasible.
> >
> > Following reviewer bVqH’s suggestion, we asked annotators not to **rank or rate the methods**, but to **mark inconsistent regions** in the generated images relative to the reference. This approach directly supports the construction of the oracle metric described in Section 4.2.
> >
> > Prior to this, we conducted a **small-scale pilot study** where annotators rated generated images on a **1–100 scale**, similar to the VLM evaluation setup. Interestingly, all models received **very similar average scores (~70/100)**, despite clear visual differences between their outputs. This highlighted the irrelevance of this ungrounded scoring approach in studying subject visual inconsistency.
> >
> > We therefore greatly appreciated reviewer bVqH’s recommendation to shift to **region-based annotation**, which provided a **more precise and interpretable measure** of visual inconsistency and enabled meaningful correlation analysis across metrics.
> >
> > As this is a new task and benchmark, one of the key takeaways from our process is that **asking users to localize inconsistencies yields more actionable and reliable data** than asking them to rate overall quality.

---

> > > ### Comment · Reviewer_SfLC · 2025-08-04
> > >
> > > Thanks to the authors for the response.
> > > While I still have some concerns about the sensitivity of the features (e.g., in the last two rows of figure 3, when the pose changes significantly, VSM score is much lower), the authors have addressed most of my concerns.
> > >
> > > The additional experiments and user study show that the features learn beyond the inpainted regions, and the paper provides new dataset and metric which can benefit the community.
> > >
> > > Therefore, I am happy to increase my score. I highly suggest the authors to add an analysis about the sensitivity of VSM to pose, illumination, scale and image degradation, that would encourage other researchers to adopt the metric.

---

### Official Review · Reviewer_bVqH · 2025-06-21

**Clarity:** 4
**Significance:** 3
**Originality:** 3
**Rating:** 5
**Confidence:** 5

**Summary:**

This paper proposes a novel metric for evaluating subject-driven image generation models. The proposed method disentangles semantic features and visual features from a frozen diffusion backbone to compute semantic similarity and visual similarity, and the final quality metric, visual-semantic match (VSM), is given by an IoU of the semantically and visually matched regions. A data generation pipeline is proposed, by altering the semantically corresponding regions using an inpainting model. Inconsistent regions can be localized.

**Questions:**

Following up from the Weaknesses above:
1. It would be good to show visualizations of semantic and visual correspondences on samples with large view/pose change and lighting difference. Is the proposed metric robust to such variations? What if the object gets heavily occluded after the subject-driven generation? Would this result in a lower metric value?
2. A more objective way to evaluate the metrics would be to actually generate results using subject-driven generation models and have human annotators manually label the inconsistent regions against the reference object images, even on just a small number of samples. The oracle metric can be computed on such human annotations to evaluate the metrics.

This is a great paper proposing an evaluation metric for subject-driven generation, with promises to release the code and model. I'm more closer to Accept and willing to raise the score based on the responses to #1 and #2.

**Ethical Concerns:**

["NO or VERY MINOR ethics concerns only"]

**Final Justification:**

Great discussion with the authors during the discussion period. The authors were able to clarify all my questions and concerns the best they can given that no visual results could be provided. I trust the authors on the updated visual results and to make appropriate revisions as promised. The additional results provided during the discussion should go in the main paper or the supplementary material as they seem fit. Happy to raise my score and recommend Accept for this paper. I believe the proposed metric will be able to serve as a reliable quality metric for evaluating subject-driven generation models, where such metric was highly needed.

**Limitations:**

Limitations have been discussed. Potential negative societal impact has not been discussed.

**Quality:**

3

**Strengths And Weaknesses:**

Strengths:
* It has long been a challenge in subject-driven image generation that there are no reliable quality metrics to evaluate models. This paper addresses this well and can potentially have a big impact in the domain.
* The whole pipeline is sound, from data generation using a diffusion-based inpainting model and training the model via localized contrastive learning.
* Visualizations of inconsistent regions look accurate. The proposed metric (VSM) works much better than CLIP, DINO or VLM, which are metrics that are currently being used in the domain.
* Various ablation studies are provided in the main paper and the supplementary material, allowing readers to gain a deeper understanding of the proposed approach.
* The authors promise to publicly release the source code and the proposed model.
* The paper is well-written. It is clear and easy to read with motivations explained well throughout.

Weaknesses:
* My main concern is that the visualized samples in Fig 1 (main), Fig 6 (main) and Fig 2 (supp) mostly show cases where there is almost no view change of the reference object. For evaluating subject-driven generation results, I think it's important that the metric is invariant to view and lighting changes of the object and solely focuses on the identity.
* Metrics are evaluated using the proposed data generation pipeline in Table 1, in which case, the proposed model could be at an advantage regardless of the specific images being held out. Metrics should be evaluated on real results by subject-driven generation models for a more objective and accurate evaluation. A suggestion is given in Questions #2 below.

---

> ### Author Rebuttal · Authors · 2025-07-29
>
> &emsp; We thank the reviewer for the constructive and thoughtful feedback, and for highlighting several strengths of our work, including that **“this is a great paper proposing an evaluation metric for subject-driven generation,”** that it **“can potentially have a big impact in the domain,”** that **“the whole pipeline is sound,”** and that **“visualizations of inconsistent regions look accurate.”**
> We also appreciate the recognition of the paper’s clarity and the effectiveness of the proposed VSM metric compared to existing alternatives such as CLIP, DINO, and VLM.
>
> We address the reviewer’s concerns and suggestions below:
>
> ---
>
> > **“My main concern is that the visualized samples in Fig 1 (main), Fig 6 (main) and Fig 2 (supp) mostly show cases where there is almost no view change of the reference object. For evaluating subject-driven generation results, I think it's important that the metric is invariant to view and lighting changes of the object and solely focuses on the identity.”**
>
> &emsp;  We agree with the reviewer that the qualitative examples in Figures 1 and 6 of the main paper do not include cases with significant pose or lighting variation.
> We updated Figure 6 to include examples with large pose and view variations.
> However, due to recent changes in the NeurIPS rebuttal policy, we are unable to include additional visual examples at this stage.
>  **Therefore, we will include the updated figure in the revised version of the paper**.
>
> In the meantime, we would like to draw the reviewer’s attention to **Figure 3 in the supplementary material**, which includes examples with **notable changes in pose and illumination** from real subject-driven generation outputs.
> These examples demonstrate that our method **is effective at localizing inconsistent regions**, even under changes of pose and viewpoints.
>
> ---
>
> > **“Metrics are evaluated using the proposed data generation pipeline in Table 1, in which case, the proposed model could be at an advantage regardless of the specific images being held out. Metrics should be evaluated on real results by subject-driven generation models for a more objective and accurate evaluation. A suggestion is given in Questions #2 below.”**
>
> &emsp; We agree with the reviewer that evaluating metrics solely on data generated by our own pipeline may not fully reflect real-world performance. To address this, we conducted an additional evaluation on **real subject-driven generation outputs**, as described in **Section D of the supplementary material**.
>
> In this evaluation, we applied our VSM metric to outputs from three recent subject-driven generation models: **Diptych Prompting**, **DSD Diffusion**, and **EasyControl**. These examples exhibit natural visual inconsistencies and do not involve any synthetic inpainting. Results show that our method effectively localizes inconsistent regions and assigns **more reasonable and differentiated scores** to each method. In contrast, existing metrics such as CLIP, DINO, and VLM often produce **similar or inaccurate scores**, even when inconsistencies are clearly visible.
>
> This real-world evaluation highlights the **robustness and practical utility** of our metric beyond synthetic settings.
>
> We address the follow-up suggestion in **Question #2** separately below.
>
> ---
>
> > **Q1 “It would be good to show visualizations of semantic and visual correspondences on samples with large view/pose change and lighting difference. Is the proposed metric robust to such variations? What if the object gets heavily occluded after the subject-driven generation? Would this result in a lower metric value?”**
>
> &emsp; As discussed above, we will include additional visualizations of semantic and visual correspondences under large pose and lighting variations in the revised version of the paper.
>
> Regarding the robustness of the metric in the presence of occlusion, the VSM metric compares visual correspondences against semantic ones, using an IoU-style formulation over matched point correspondences.
> If a region becomes heavily occluded in one of the images, it is **unlikely to be selected as a semantic match**.
> As a result, such regions would typically be excluded from the semantic match set, and thus **do not contribute to the VSM score**.
> This behavior aligns with human judgment: VSM focuses on regions that are both semantically and visually present in both views, and naturally disregards parts that are missing or occluded in one image.
>
> This design makes the metric **robust to occlusion and partial visibility**, ensuring that the score reflects consistency only over **mutually visible parts** of the subject.
>
> ---
>
> > **Q2 “A more objective way to evaluate the metrics would be to actually generate results using subject-driven generation models and have human annotators manually label the inconsistent regions against the reference object images, even on just a small number of samples. The oracle metric can be computed on such human annotations to evaluate the metrics.”**
>
> This is an excellent suggestion, and we thank the reviewer for proposing it.
>
> To implement this idea, we developed a lightweight web-based annotation tool where the **reference image** is displayed on the left and the **generated image** from a subject-driven model is shown on the right. Annotators are asked to draw masks over regions in the generated image that are **visually inconsistent** with the reference subject.
>
> To compute the oracle score, we follow the protocol described in Section 4.2, calculating the **ratio between the annotated inconsistent region and the reference object mask**, providing a normalized inconsistency score per sample.
>
> We asked professional annotators to label a subset of generations from our test set, resulting in 480 high-quality annotated samples. We then computed the correlation between each metric and this human-annotated oracle:
>
> | Metric         | Pearson Correlation | Spearman Correlation |
> |----------------|---------------------|-----------------------|
> | CLIP           | 0.223               | 0.172                 |
> | DINOv2         | 0.107               | 0.132                 |
> | VLM (GPT-4o)   | 0.298               | 0.147                 |
> | **VSM (Ours)** | **0.367**           | **0.253**             |
>
> Our **VSM metric consistently achieves the highest correlation** with human annotations, demonstrating its strong performance on real-world data. While VLM (GPT-4o) performs better in this setting than in the synthetic experiments reported in the main paper, our approach still maintains the **highest overall correlation**, confirming its robustness and reliability.
>
> This experiment further validates that our approach:
> - Generalizes beyond the synthetic training setup,
> - Produces objective and interpretable consistency scores,
> - And uniquely supports **localized, pixel-level consistency maps**, a capability not provided by current metrics or VLMs.
>
> Together, these strengths position VSM as a **practical and reliable evaluation tool** for subject-driven generation tasks.

---

> ### Comment · Reviewer_bVqH · 2025-08-04
>
> Thanks for updating Figure 6 and pointing to Section D. Q1 is clarified and the robustness to occlusion seems like another advantage of the proposed metric. The additional evaluation in Q2 is much appreciated – it clearly shows that the proposed metric is the most reliable (i.e., most highly correlated to human annotations). I am confident to raise my score to Accept.
>
> Additional question after looking at Figure 3 in the supp:
> * Could there be a bias in the proposed metric to favor results with less view change even if the identity is preserved? An extreme way to put it is, would copy-and-pasting the object get a very high score? For example, in the yellow tractor case in Figure 3, I felt the front yellow part is more consistent in EasyControl vs Diptych (EasyControl has larger view change), however EasyControl has a much lower score. I think a systematic way to show the robustness to view change is to measure the proposed metric on multi-view datasets (e.g., MVImgNet) with ground truth multi-view images and show that the metric value is more-or-less consistent. The same can be done for lighting changes (if relighting is hard, I guess it's always possible to randomly perturb the color and try measuring the proposed metric to see if it's more-or-less consistent).
>
> Additional question from the response to Q1:
> * I assume newly generated areas of the object (e.g., due to big view changes) may have no semantic matches to the reference image, similar to occluded regions, and will not contribute to the proposed metric. Do the authors have any idea how to measure the consistency or naturalness of such newly generated regions? I understand this may not be the scope of this paper.
>
> I am noting that these two additional questions are optional, for more discussion on the paper.

---

> > ### Author Response · Authors · 2025-08-04
> >
> > We thank the reviewer for their response and for raising the score to Accept.
> >
> > ---
> > > **“Could there be a bias in the proposed metric to favor results with less view change even if the identity is preserved? An extreme way to put it is, would copy-and-pasting the object get a very high score? For example, in the yellow tractor case in Figure 3, I felt the front yellow part is more consistent in EasyControl vs Diptych (EasyControl has larger view change), however EasyControl has a much lower score.**
> >
> > At first glance, **EasyControl** may appear more consistent due to producing **higher-resolution and sharper images**, as it operates at the full FLUX resolution. In contrast, **Diptych** and **DSD** use in-context generation approaches at half the resolution, which can result in softer visual output.
> >
> > However, upon closer inspection, **EasyControl tends to simplify or alter fine-grained details**, even when overall sharpness is higher. In the **yellow tractor example** (Figure 3), several key inconsistencies in the front yellow part of EasyControl’s output contribute to its lower VSM score compared to Diptych:
> >
> > - The **upper cylinder** on the front yellow part is **black** in the reference image, but generated as **yellow** in EasyControl.
> > - The **middle cylinder** is **carved** in the reference, but appears **smooth** in EasyControl.
> > - The **bolts** are **yellow** in the reference, but **silver** in EasyControl.
> > - The **bucket geometry** differs: it has **curved angles** in the reference, but is **straightened** in EasyControl.
> >
> > These subtle inconsistencies, especially in color, texture, and geometry, are captured by our **VSM metric**, which evaluates consistency at the **visual feature level**, beyond just resolution or apparent sharpness.
> >
> > ---
> >
> > > **“I think a systematic way to show the robustness to view change is to measure the proposed metric on multi-view datasets (e.g., MVImgNet) with ground truth multi-view images and show that the metric value is more-or-less consistent. The same can be done for lighting changes (if relighting is hard, I guess it's always possible to randomly perturb the color and try measuring the proposed metric to see if it's more-or-less consistent).”**
> >
> > That is an excellent suggestion, and we thank the reviewer for proposing it.
> > In the **revised version of the paper**, we will include a **sensitivity analysis** that evaluates the robustness of our metric under controlled variations, including **pose, illumination, and blur**. This will provide deeper insights into the stability of the VSM metric under real-world changes.
> > This analysis will help demonstrate the **view- and lighting-invariance** of the learned visual features and the **reliability of the VSM score** under diverse conditions.
> >
> > ---
> >
> > > **“I assume newly generated areas of the object (e.g., due to big view changes) may have no semantic matches to the reference image, similar to occluded regions, and will not contribute to the proposed metric. Do the authors have any idea how to measure the consistency or naturalness of such newly generated regions? I understand this may not be the scope of this paper.”**
> >
> > In case of complete view changes, i.e., 90 degrees rotation of the object. There are a few semantic matches that are matched, and therefore, the visual consistency can not be effectively measured.
> >
> > Addressing this challenge is a compelling direction for future work, but it remains non-trivial, as it involves comparing two nearly disjoint sets of regions with little to no direct overlap. Evaluating the plausibility or coherence of newly generated content in the absence of visual matches may require fundamentally different strategies, such as 3D reconstruction, pose estimation, or lifting the problem into 3D space and treating it as a 3D generation and evaluation task.

---

> > > ### Comment · Reviewer_bVqH · 2025-08-04
> > >
> > > Appreciate the additional discussion. I don't think this part was answered:
> > > > **Could there be a bias in the proposed metric to favor results with less view change even if the identity is preserved? An extreme way to put it is, would copy-and-pasting the object get a very high score?**
> > >
> > > For the sensitivity analysis, it would be helpful if the authors can provide any preliminary results before the discussion period ends so that reviewers can factor it in during the final discussion phase among reviewers. The easiest way would be to try measuring the metric on multi-view images in MVImgNet or any other multi-view dataset (just a small subset is fine for a preliminary analysis). There's not much time so this is optional but would be much appreciated.

---

> > > > ### Author Response · Authors · 2025-08-05
> > > >
> > > > We apologize for missing this part.
> > > >
> > > > > **"Could there be a bias in the proposed metric to favor results with less view change even if the identity is preserved? An extreme way to put it is, would copy-and-pasting the object get a very high score?"**
> > > >
> > > > We did not observe any systematic bias in our metric toward favoring results with smaller view changes. While larger view changes naturally reduce the number of matched regions, unmatched regions do not affect the metric, as explained earlier in the context of occlusion.
> > > >
> > > > Regarding the copy-and-paste scenario, since our metric is computed based on correspondences, it will not benefit from copy-and-paste operations and remains agnostic to spatial alignment.
> > > >
> > > > ---
> > > >
> > > > We will start with the sensitivity analysis and will do our best to post some preliminary results during the discussion phase.

---

> ### Author Response · Authors · 2025-08-09
>
> Following up on Reviewer bVqH’s suggestion, we began performing a sensitivity analysis of our approach.
> We initially attempted to download **MVImgNet** for this purpose, as suggested by the reviewer, but encountered persistent issues with the download server, resulting in repeated disconnections.
>
> This led us to a more suitable alternative: the **DreamBooth benchmark**, which is directly relevant to our setting. DreamBooth contains real images of the same subject captured under **different poses, lighting conditions, and even non-rigid deformations**, enabling us to evaluate the robustness of VSM across a wide range of environmental and subject variations.
>
> For this study, we selected **8 subjects** from DreamBooth:
> `backpack`, `shiny_sneaker`, `duck_toy`, `wolf_plushie`, `robot_toy`, `rc_car`, `bear_plushie`, `monster_toy`.
> We refer reviewers to the official DreamBooth repository for visual examples of these subjects.
>
> For each subject, we chose **two images** with **significantly different poses** and, in most cases, also differing in **lighting** and **non-rigid deformations**. We manually annotated regions in the image pairs and inpainted them to create controlled inconsistencies. The **oracle** was then computed following the protocol described in Section 4.2.
> Note that we do not train or fine-tune our approach on this DreamBooth subset, but we use the checkpoint pre-trained on our automatically generated dataset.
>
> The correlation between each metric and the oracle on this set is reported below:
>
> | Metric              | Pearson Correlation | Spearman Correlation |
> |---------------------|---------------------|----------------------|
> | CLIP                | -0.363              | -0.309               |
> | DINOv2              | 0.453               | 0.047                |
> | VLM (ChatGPT-4o)    | 0.462               | 0.185                |
> | **VSM (Ours)**      | **0.846**           | **0.431**            |
>
> These preliminary results show that **VSM consistently outperforms all other metrics**, even under substantial **pose, lighting, and deformation changes**.
> We also observe that all metrics perform better on this dataset compared to the test set in **Table 1** of the main paper, which we attribute to the smaller size of this DreamBooth subset (8 samples, compared to 120 samples in the original test set).
>
> To strengthen these findings, we plan to extend this analysis by **expanding the dataset** and **isolating different variation factors**, such as pose, lighting, and non-rigid deformation, to evaluate VSM’s robustness to each independently and more systematically.
>
> We will include these findings in the revised version of the paper.

---

### Official Review · Reviewer_oXtL · 2025-07-01

**Clarity:** 3
**Significance:** 2
**Originality:** 2
**Rating:** 4
**Confidence:** 3

**Summary:**

This paper proposes a method to evaluate visual consistency in subject-driven image generation by disentangling semantic and visual features from diffusion model backbones. The approach includes an automated dataset generation pipeline that creates image pairs with visual inconsistencies, training aggregation networks with a contrastive loss to separate semantic from visual features, and a metric based on the trained model for quantifying consistency. The metric shows improved correlation with ground-truth consistency compared to existing metrics like CLIP and DINO, while also providing spatial localization of inconsistent regions.

**Questions:**

Have you considered fine-tuning an open source VLM with this data? Given the detailed information about inconsistency locations, this could be effective even with limited training steps.
Why do you use features from diffusion UNets rather than Transformer models - is this to leverage previous studies on UNet features or for other reasons?

**Ethical Concerns:**

["NO or VERY MINOR ethics concerns only"]

**Final Justification:**

Most of my concerns were address in the rebuttal, and having read the discussions with other reviewers, I maintain my original positive leaning score.

**Limitations:**

yes

**Quality:**

2

**Strengths And Weaknesses:**

Strengths:

(1) Well-motivated approach: The work addresses two relevant gaps: (a) A metric for subject consistency is needed, as existing approaches like CLIP and DINO focus on global semantics and may miss visual details, and (b) as noted in the paper, diffusion features have been used extensively for semantic correspondence and other tasks, but less so for subject consistency evaluation.
(2) Useful data generation pipeline: The automated pipeline for creating image pairs with controlled visual inconsistencies could be valuable regardless of the downstream evaluation method. Even if VLM-based approaches prove better than pretrained diffusion features for consistency evaluation, this synthetic data could be used to fine-tune VLMs.


Weaknesses:

(1) VLM-based metrics argument: The critique of VLM-based approaches may be overstated. Models like Gemini have shown reasonable capability in localizing visual inconsistencies, and the limitations noted could potentially be addressed through improved prompting strategies. It's also unclear why a single consistency score is preferable to region-based analysis that VLMs could provide.
(2) ”Disentanglement" terminology: The term "disentanglement" seems overstated given that the primary goal is subject consistency evaluation rather than true feature disentanglement. The learned visual features likely still contain semantic information - for instance, the method probably wouldn't identify two images with identical texture but completely different semantic content as the same subject. The work might be better framed around "subject-driven features" or similar terminology that better reflects the actual objective.
(3) Missing citations: The related work appears to miss relevant approaches in training-free consistent generation, including [1] ConsiStory: Training-Free Consistent Text-to-Image Generation [2] The Chosen One: Consistent Characters in Text-to-Image Diffusion Models (both appeared in SIGGRPAH 2024)

---

> ### Author Rebuttal · Authors · 2025-07-29
>
> We thank the reviewer for the constructive feedback and thoughtful suggestions, as well as for recognizing the strengths of our approach, including that it presents a **“well-motivated approach,”** **“addresses two relevant gaps,”** and offers a **“useful data generation pipeline”** whose **“synthetic data could be used to fine-tune VLMs.”**
>
> We address the reviewer’s questions and concerns point by point below.
>
> ---
>
> > **(1) “VLM-based metrics argument: The critique of VLM-based approaches may be overstated. Models like Gemini have shown reasonable capability in localizing visual inconsistencies, and the limitations noted could potentially be addressed through improved prompting strategies. It's also unclear why a single consistency score is preferable to region-based analysis that VLMs could provide.”**
>
> &emsp; We agree that VLM-based approaches can provide valuable insights for evaluating subject-driven generation, particularly in delivering high-level judgments of visual consistency, as demonstrated in DreamBench++.
>
> However, as shown in **Figure 3 of the supplementary material**, VLMs, including ChatGPT-4o, often produce **overly optimistic or indistinguishable scores across different generation methods**, even in the presence of clear visual inconsistencies. This highlights the limitations of relying solely on language-conditioned scoring for detecting fine-grained visual differences.
>
> Regarding the use of Gemini, we share the reviewer’s view that it has strong potential, especially given its native grounding and multimodal segmentation capabilities. However, to the best of our knowledge, there is currently **no published work** evaluating Gemini’s ability to localize inconsistencies in subject-driven generation.
>
> To explore this direction, we conducted **preliminary experiments** in which we concatenated the reference image and the generated image side-by-side and prompted Gemini to produce segmentation masks highlighting inconsistent regions. Unfortunately, we were **unable to obtain valid outputs via the API**, suggesting either prompt misalignment or unsupported task formulation.
>
> We still believe Gemini may be a promising direction, but several open design questions remain:
> - Should images be presented independently or concatenated?
> - Can the model return pixel-level consistency scores or only segmentation masks?
> - What prompt formulations are required to elicit meaningful pairwise comparisons?
>
> Without a systematic investigation into these questions or supporting literature, we believe it is **premature to draw conclusions** about Gemini's applicability to this task. That said, we welcome any references or suggestions from the reviewer regarding published work in this area.
>
> By contrast, our method already provides a unified and transparent solution offering:
> - A **global consistency score** via the VSM metric
> - **Localized, pixel-level consistency maps**, as demonstrated in **Figure 3** in the supplementary.
>
> This combination of **quantitative scoring and spatial interpretability** remains **unmatched by current VLM-based metrics**, and we believe it fills a critical gap in the evaluation of subject-driven generation.
>
>
> ---
>
> > **(2) “Disentanglement" terminology: The term "disentanglement" seems overstated given that the primary goal is subject consistency evaluation rather than true feature disentanglement. The learned visual features likely still contain semantic information — for instance, the method probably wouldn't identify two images with identical texture but completely different semantic content as the same subject. The work might be better framed around "subject-driven features" or similar terminology that better reflects the actual objective.**
>
> &emsp; We appreciate this insightful observation and agree with the reviewer that the learned visual features likely retain some degree of semantic information, particularly because the training image pairs consist of the **same object**, even after visual perturbation.
>
> Achieving complete disentanglement, where visual features are entirely free of semantic content, would ideally require a dataset that provides **visual and semantic correspondences across different objects**, a setup that is not straightforward to construct in an automated way and would require huge manual annotation effort.
>
> Therefore, we leave this to future work, and we will revise the manuscript to clarify this point and **acknowledge that our notion of "disentanglement" is partial and task-driven**.
> We will also incorporate this discussion in the **limitations and future work sections** of the revised paper.
>
> ---
>
> > **(3) “Missing citations:” The related work appears to miss relevant approaches in training-free consistent generation, including [1] ConsiStory: Training-Free Consistent Text-to-Image Generation [2] The Chosen One: Consistent Characters in Text-to-Image Diffusion Models (both appeared in SIGGRPAH 2024)**
>
> &emsp; We thank the reviewer for pointing that out. We will include the missing references in the revised manuscript.
>
> ---
>
> > **“Have you considered fine-tuning an open source VLM with this data? Given the detailed information about inconsistency locations, this could be effective even with limited training steps.”**
>
> &emsp; We appreciate this excellent suggestion. Our automated dataset generation pipeline indeed provides detailed annotations of inconsistent regions, which could be valuable for fine-tuning vision-language models to improve their ability to detect visual inconsistencies.
>
> While promising, such an approach would require addressing several design challenges, including how to represent the annotated regions as input to the VLM (e.g., through bounding boxes, masks, or spatial tokens), and what **output modality** the VLM should be trained to produce (e.g., region-wise judgments or pixel-level scores).
>
> We view this as a compelling direction and will include this idea as part of the **future work** in the revised manuscript.
>
> ---
>
>
> > **“Why do you use features from diffusion UNets rather than Transformer models — is this to leverage previous studies on UNet features or for other reasons?”**
>
> Yes, our choice to use **UNet-based diffusion backbones** was driven by both practical and methodological considerations. UNet backbones have been extensively studied in prior work, and with the emergence of **CleanDIFT**, it became possible to extract clean features with *near real-time efficiency*. This enables fast and scalable computation of dense correspondences, which our VSM metric relies on, with minimal overhead.
>
> That said, we found a promising new work [1] that explores using **DiT-based backbones** for semantic correspondence tasks. This work appeared **after the NeurIPS submission deadline**, and we were therefore unable to consider it in our method or comparisons. Once the code and model checkpoints become available, we are keen to explore integrating DiT-based features into our framework.
>
> Importantly, our pipeline is **modular and backbone-agnostic by design**, and can readily incorporate new backbones, including DiT, as the field matures and extraction tools become more accessible.
>
> [1] Gan, Chaofan, et al. "Unleashing Diffusion Transformers for Visual Correspondence by Modulating Massive Activations." arXiv preprint arXiv:2505.18584 (2025).

---

### Official Review · Reviewer_Ke5D · 2025-07-08

**Clarity:** 3
**Significance:** 2
**Originality:** 3
**Rating:** 4
**Confidence:** 4

**Summary:**

A major bottleneck in the subject-driven generation research line is the lack of reliable evaluation metrics for subject consistency. The traditional similarity metrics such as LPIPS or structural similarity (SSIM) are not well suited for this task as subjects may appear in varying poses and spatial configurations. Feature-based similarities CLIP or DINO score does not work well either as they overlook subtle visual inconsistencies in object details. This work propose a novel approach for disentangling visual and semantic features, enabling the detection of visually inconsistent regions in subject-driven image generation. To achieve such goal, they first introduce an automated pipleine for constructing a dataset with visual correspondences, leveraging existing datasets of subject-driven image generation such as Subjects200k. Then they propose an architecture for disentangling semantic and visual representations from the internal features of diffusion models. Lastly, they introduce a metric that leverages the disentangled features for empirically evaluating visual consistency between image pairs, which quantifies the degree of consistency and localizes inconsistent regions. Empirical results show that their methods achieves SOTA correlation score with oracle compared to other feature-based similarity metrics and VLM judgement.

**Questions:**

- Can you offer more results with large pose variations and non-rigid deformations?
- During the inpainting process, do you see any artifects such as the clear boundary of the inpainting region? If yes, is it an obvious image cue for the network to capture during the training process and hack the final test score?
- Is there a multiple visual inconsistence setup in the experiment? In the text, it only says the smallest region for each SAM segmentation to create the inpainting. Since in reality, there always are multiple inconsistencies in the results, how to evaluate such senarios?

**Ethical Concerns:**

["NO or VERY MINOR ethics concerns only"]

**Final Justification:**

As authors have well addressed my major concerns, and I have read all the responses to other reviewers, I would like to lift the final rating of the proposal.

**Paper Formatting Concerns:**

yes

**Quality:**

3

**Strengths And Weaknesses:**

Strengths:
- It is the first approach that tackles both the semantic and visual inconsistency in subject-driven image generation. The proposal includes the dataset creation, baseline method built-up, and evalution metics, so it is a complete work.
- The design details of data pipeline is clever. Fexible utilization of grounded-sam, sam, skewness score, inpainting model and LPIPS score guarantees the quality of the dataset.
- The proposed method shows significant superiority against the competitors on their own dataset, creating a good baseline solution of this topic.

Weakness:
- The novelty of the solution in learning disentangled semantic and visual representation is low.  The features are computed from a frozen diffusion network in a dual-stream design, and the learning process is to determine how to aggregate the importance of those features from different layers of diffusion model. And the work only investigates SD 2.1 structure and no other popular DiT based networks.
- The variations of objects, poses, illumination, background, etc. in the dataset is limited due to the only data source from Subjects 200K. For example, in the qualitative exmaples offered in figure 6, we didn't see much pose variations or any other non-rigid deformations of objects. Whether the network can maintain a good semantic similarity under large variations is a question, though it can captures small visual diffierences.

---

> ### Author Rebuttal · Authors · 2025-07-29
>
> We thank the reviewer for the detailed and thoughtful feedback, and for positively highlighting key aspects of our work,  including that it is **“the first approach that tackles both the semantic and visual inconsistency,”** that our **“data pipeline is clever,”** and that our method shows **“significant superiority against the competitors.”** We greatly appreciate the recognition of the novelty and completeness of our pipeline.
>
> We address the reviewer’s concerns and questions one by one below:
>
> ---
>
> > **“The novelty of the solution in learning disentangled semantic and visual representation is low. The features are computed from a frozen diffusion network in a dual-stream design, and the learning process is to determine how to aggregate the importance of those features from different layers of diffusion model.”**
>
> &emsp; While our architecture indeed builds on the principle of feature aggregation, a widely used strategy in analyzing high-dimensional networks such as diffusion models and DINO [1,2], our contribution lies not in the aggregation mechanism, but in developing a **comprehensive and first-of-its-kind framework** for disentangling **semantic and visual representations** from diffusion model backbones.
>
> As the reviewer rightly points out, our work is the first to explicitly address the challenge of **visual correspondence** within diffusion models, complementing the well-studied semantic correspondence task, with the aim of detecting visual inconsistencies in subject-driven generation. To tackle this novel task, we introduce several key contributions:
>
> - A **carefully designed dataset generation pipeline**, which combines established tools (e.g., SAM, LPIPS) with a novel **skewness-based metric** to ensure high-quality correspondences.
>
> - A **dual-branch aggregation architecture**, trained with a contrastive objective to disentangle visual and semantic features; a setup that was not investigated before.
>
> - A new **Visual-Semantic Match (VSM)** metric that enables **both quantification and spatial localization** of inconsistencies, capabilities absent in current evaluation protocols.
>
> In summary, while the individual building blocks (e.g., aggregation, feature alignment) are familiar, our work makes a novel and cohesive **methodological contribution** to a previously unexplored problem setting, which we believe is a necessary foundation for future research in visual consistency evaluation.
>
> ---
>
>
>
>
>
>
>
> > **“The work only investigates SD 2.1 structure and no other popular DiT based networks.”**
>
> &emsp; Despite the increasing prominence of DiT-based diffusion models in image generation, efficient feature extraction from their backbones remains an active area of research, particularly when it comes to identifying which layers yield robust semantic or visual features [3].
>
> Moreover, DiT architectures are computationally expensive, which makes dense feature extraction and aggregation significantly slower than UNet-based models. In contrast, UNet-based diffusion models benefit from well-studied internal representations and recent advancements such as **CleanDIFT** [4], which distills clean features from intermediate layers and enables *near* real-time inference for correspondence tasks.
>
> That said, we would like to acknowledge that we found a recent preprint [5] addressing the use of DiT-based diffusion backbones for semantic correspondence. This paper was made publicly available on arXiv **after the NeurIPS submission deadline**, and we were therefore unable to consider it during the development and writing of our submission. Once the **code and implementation details become available**, we are very interested in integrating DiT features into our pipeline and evaluating their impact on visual correspondence tasks.
>
> In summary, while DiTs show strong generative potential, their internal representations are still being mapped and benchmarked. Our work builds on well-established UNet backbones for practical and reproducibility reasons, but is structured to **adopt DiT backbones as the field matures**.
>
> ---
>
> > **“The variations of objects, poses, illumination, background, etc. in the dataset is limited due to the only data source from Subjects 200K … Whether the network can maintain a good semantic similarity under large variations is a question, though it can captures small visual diffierences”**
>
> &emsp; We agree with the reviewer’s concern that relying solely on a single dataset can impose limitations in terms of object pose diversity, illumination conditions, and background variations. However, at the time of submission, **Subjects200K was the only publicly available curated dataset** for subject-driven generation.
>
> That said, the **qualitative results provided in Section D and Figure 3 of the supplementary material**, where we evaluate multiple subject-driven generation methods using our proposed metric, **demonstrate that our approach is effective even under large pose changes** in the generated subjects.
>
> Finally, we emphasize that our **dataset generation pipeline is highly flexible** and can be applied to future subject-driven generation datasets as they emerge. Since it only requires **image pairs of the same subject**, it imposes minimal assumptions, ensuring **broad applicability beyond the current dataset** and paving the way for seamless extension to more diverse settings.
>
> ---
>
>
>
> > **“Can you offer more results with large pose variations and non-rigid deformations?”**
>
> We have updated **Figure 6** to include additional examples that exhibit **large pose variations and non-rigid deformations**. However, due to the recent changes in NeurIPS rebuttal policy, we are unable to upload revised figures or supplementary content during the rebuttal phase.
>
> These updated examples will be **included in the revised version of the paper** to better demonstrate the robustness of our method under more challenging transformations.
>
> ---
>
> > **“During the inpainting process, do you see any artifacts such as the clear boundary of the inpainting region? If yes, is it an obvious image cue for the network to capture during the training process and hack the final test score?”**
>
> &emsp; This is an important concern and one we explicitly considered when designing our inpainting pipeline. To minimize the risk of the model learning to detect the inpainting artifact, we carefully tuned the inpainting pipeline to minimize introducing visible seams or hard edges. Specifically, we **crop a localized patch around the region to be inpainted with padding**, which promotes smoother blending and reduces the likelihood of visible boundaries.
>
> Moreover, since our method operates on **point-based correspondences rather than region-level supervision**, the majority of training points lie **within the inpainted region itself**, away from boundaries where artifacts might appear.
>
> To further rule out the possibility that the model is simply learning to detect inpainting artifacts, we evaluated it on **real subject-driven generation outputs** in **Section D and Figure 3 of the supplementary material**, where no synthetic inpainting is involved. In these real-world scenarios, our method continues to **successfully localize visual inconsistencies**, while competing metrics such as CLIP, DINO, and VLM struggle to differentiate outputs.
>
> These results strongly suggest that our learned features do **not rely on artifact cues**, and instead generalize to meaningful appearance differences across naturally generated image pairs.
>
> ---
>
> > **“Is there a multiple visual inconsistence setup in the experiment? In the text, it only says the smallest region for each SAM segmentation to create the inpainting. Since in reality, there always are multiple inconsistencies in the results, how to evaluate such scenarios?”**
>
> &emsp; This is a valid concern and one we considered in the design of our dataset generation pipeline.
>
> While we select the **smallest valid region** from SAM segmentation during inpainting to ensure precision and control, it is important to note that **our training pipeline operates at the level of point correspondences, not regions**.
> As a result, the learning process is inherently **agnostic to the number of inconsistent regions**, whether one or many, since the contrastive objective is applied over **individual point correspondences**.
>
> In practice, as shown in **Figure 3 of the supplementary material**, our approach is capable of successfully localizing **multiple** inconsistent regions in the generated subject, for example, the front and rear wheels of the toy tricycle, or the handle and blade of the rolling cutter.
>
> These results confirm that, even though each training sample may contain only a single manipulated region, the model generalizes well to real-world cases involving multiple inconsistent regions, which are common in subject-driven generation outputs.
>
> ---
>
> **References**
> [1] Luo, Grace, et al. "Diffusion hyperfeatures: Searching through time and space for semantic correspondence." *NeurIPS* (2023)
>
> [2] Zhang, Junyi, et al. "Telling left from right: Identifying geometry-aware semantic correspondence." *CVPR* (2024)
>
> [3] Avrahami, Omri, et al. "Stable flow: Vital layers for training-free image editing." CVPR (2025)
>
> [4] Stracke, Nick, et al. "Cleandift: Diffusion features without noise." CVPR (2025)
>
> [5] Gan, Chaofan, et al. "Unleashing Diffusion Transformers for Visual Correspondence by Modulating Massive Activations." arXiv preprint arXiv:2505.18584 (2025).

---

> > ### Author Response · Authors · 2025-08-07
> >
> > Dear Reviewer,
> >
> > Thank you again for your thoughtful feedback and constructive suggestions.
> >
> > In our rebuttal, we made a concerted effort to address all of your concerns in detail, and we believe your comments have significantly helped us improve the clarity and completeness of the paper.
> >
> > If you have any further questions or additional suggestions that could help strengthen the work, we would greatly appreciate your input.
> >
> > We also kindly ask you to consider reassessing your score in light of the clarifications and updates provided.

---

> > ### Comment · Reviewer_Ke5D · 2025-08-07
> >
> > Thank you for the detailed response.
> > My major concerns about the novelty is well addressed in the reponse. I would suggest authors re-organize the text a bit to highlight the methodological contributions in the revision.
> >
> >  It is a bit disappointed that authors do not offer a quantitative analysis for the cases with large pose variations due to revision restrictions. I think some basic statistics are enough to show whether the network ability is constrained by the pose variations.I hope to see the results from the updated figure in the future revision.

---

> > > ### Author Response · Authors · 2025-08-09
> > >
> > > We are glad that the reviewer’s **concerns about the novelty have been well addressed**.
> > > In the revised version, we will **re-organize the text** to more prominently highlight the methodological contributions.
> > >
> > > We understand the disappointment at not being able to provide additional qualitative figures during the rebuttal phase. We have updated **Figure 6** to include samples with **higher pose variations**, and this will be included in the revised version.

---

> > > > ### Author Response · Authors · 2025-08-09
> > > >
> > > > Following up on whether the network's ability is constrained by the pose variations, we conducted a preliminary study to evaluate the performance of our metric under large pose variations.
> > > >
> > > > We base this analysis on the **DreamBooth benchmark**, which is directly relevant to our setting.
> > > > DreamBooth contains real images of the same subject captured under **different poses, lighting conditions, and even non-rigid deformations**, enabling us to evaluate the robustness of VSM across a wide range of environmental and subject variations.
> > > >
> > > > For this study, we selected **8 subjects** from DreamBooth:
> > > > `backpack`, `shiny_sneaker`, `duck_toy`, `wolf_plushie`, `robot_toy`, `rc_car`, `bear_plushie`, `monster_toy`.
> > > > We refer reviewers to the official DreamBooth repository for visual examples of these subjects.
> > > >
> > > > For each subject, we chose **two images** with **significantly different poses** and, in most cases, also differing in **lighting** and **non-rigid deformations**. We manually annotated regions in the image pairs and inpainted them to create controlled inconsistencies. The **oracle** was then computed following the protocol described in Section 4.2.
> > > > Note that we do not train or fine-tune our approach on this DreamBooth subset, but we use the checkpoint pre-trained on our automatically generated dataset.
> > > >
> > > > The correlation between each metric and the oracle on this set is reported below:
> > > >
> > > > | Metric              | Pearson Correlation | Spearman Correlation |
> > > > |---------------------|---------------------|----------------------|
> > > > | CLIP                | -0.363              | -0.309               |
> > > > | DINOv2              | 0.453               | 0.047                |
> > > > | VLM (ChatGPT-4o)    | 0.462               | 0.185                |
> > > > | **VSM (Ours)**      | **0.846**           | **0.431**            |
> > > >
> > > > These preliminary results show that **VSM consistently outperforms all other metrics**, even under substantial **pose, lighting, and deformation changes**.
> > > > We also observe that all metrics perform better on this dataset compared to the test set in **Table 1** of the main paper, which we attribute to the smaller size of this DreamBooth subset (8 samples, compared to 120 samples in the original test set).
> > > >
> > > > To strengthen these findings, we plan to extend this analysis by **expanding the dataset** and **isolating different variation factors**, such as pose, lighting, and non-rigid deformation, to evaluate VSM’s robustness to each independently and more systematically.
> > > >
> > > > We will include these findings in the revised version of the paper.

---

### Note · Authors · 2025-08-15

We thank all reviewers for their constructive and thoughtful feedback. The questions, suggestions, and concerns raised during the review process prompted us to add new experiments, conduct additional analyses, and clarify our contributions, making the paper stronger and more complete.

**1. Novelty of the Approach (Reviewer Ke5D)**
We clarified how our contributions go beyond standard feature aggregation, emphasizing the complete framework, dataset pipeline, and metric design.
We will reorganize the text to better highlight these points.
The reviewer acknowledged that this addressed the concern.

**2. Pose Variation Examples**
Several reviewers requested more examples with large pose variations. Due to the rebuttal policy, we could not include them, but an updated **Figure 6** with such cases will appear in the revised version.

**3. Inpainting Artifact Concerns (Reviewers Ke5D, SfLC)**
We showed in **Appendix D** that our method works effectively on real subject-driven generations that include no inpainting, eliminating any concerns about artifact detection.

**4. Sensitivity to Pose, Lighting, and Deformations**
Several reviewers suggested performing a sensitivity analysis under various conditions.
We ran a preliminary study on annotated **DreamBooth** samples with large variations, showing **VSM** outperforms all baselines. We will expand this to isolate each factor.

**5. Gemini for Inconsistency Localization (Reviewer oXtL)**
Preliminary experiments we did with Gemini showed difficulty obtaining reliable pixel-level outputs.
While Gemini currently seems to be very far from accurate inconsistency localization compared to our method, we acknowledge this as a promising future.

**6. User Study on Real Generations (Reviewer bVqH)**
We followed the suggestion to collect human annotations of inconsistent regions in real generations. Results confirm that **VSM outperforms all metrics**, validating generalization beyond synthetic data.

**7. Limitations Section (Reviewer SfLC)**
We will add a dedicated **Limitations** section with the sensitivity analysis in the revised version.

**Closing Remarks**
We believe that we have addressed all major concerns with concrete experiments, clarifications, and planned revisions. The additional results confirm that **VSM is a robust, interpretable, and generalizable metric** for subject-driven generation, filling a critical gap in evaluation methodology and providing value to the community.

---

### Decision · Program_Chairs · 2025-09-17

**Decision:**

Accept (spotlight)

**Comment:**

The paper proposes a method for measuring visual consistency in subject-driven image generation by disentangling semantic and visual features from diffusion model backbones. Reviewers acknowledge several strengths of the paper, including addressing a critical gap where current metrics (CLIP, DINO) provide only global scores that miss fine-grained inconsistencies, while this approach can spatially localize where generated subjects differ from references. Additional strengths include a novel automated data generation pipeline and clear presentation. The main weaknesses noted by reviewers include limited pose/view variations in demonstrated examples and concerns about whether the method generalizes beyond its synthetic training data to real-world inconsistencies. In the rebuttal, the authors provided new experiments on subject-driven generations (DreamBooth) from real images, pointed to supplementary results demonstrating pose variations, and conducted a supportive human annotation study. All reviewers found the responses satisfactory and recommend acceptance.

The Area Chair agrees with the reviewers and recommends acceptance. The camera-ready version would benefit from including the promised additional experiments and discussing limitations more thoroughly.